# Atmosphere-Ocean-Terrain Coupled Ensemble Forecast Model Improves Forecasts of Abnormally Deflecting Tropical Cyclones

## Abstract

Deep learning-based tropical cyclone (TC) forecasting methods have demonstrated significant potential and application advantages, as they feature much lower computational cost and faster operation speed than numerical weather prediction models. However, existing deep learning methods still have key limitations: they can only process a single type of sequential trajectory data or homogeneous meteorological variables, and fail to achieve accurate forecasting of abnormal deflected TCs. To address these challenges, we present two groundbreaking contributions. First, we have constructed a multimodal and multi-source dataset named AOT-TCs for tropical cyclone forecasting in the Northwest Pacific basin. As the first dataset of its kind, it innovatively integrates heterogeneous variables from the atmosphere, ocean, and land, thus obtaining a comprehensive and information-rich meteorological dataset. Second, based on the AOT-TCs dataset, we propose a forecasting model that can handle both normal and abnormally deflected TCs. This is the first TC forecasting model to adopt an explicit atmosphere-ocean-terrain coupling architecture, enabling it to effectively capture complex interactions across physical domains. Extensive experiments on all TC cases in the Northwest Pacific from 2017 to 2024 show that our model achieves state-of-the-art performance in TC forecasting: it not only significantly improves the forecasting accuracy of normal TCs but also breaks through the technical bottleneck in forecasting abnormally deflected TCs. Project website: https://anonymous.4open.science/r/AOT-TC-111F.

## 1 Introduction

As extreme climate impacts intensify and strong El Niño events emerge Ripple et al. (2024), we are entering a new phase of dramatic climatic shifts. Recent years have witnessed repeated breaking of global climate records Terhaar et al. (2025), including sea surface temperatures (SST) and ice sheet extents. Since April 2023, global ocean surface temperatures (GOST) have remained at record levels for consecutive months, with the April 2023–March 2024 average exceeding the 2015–2016 historical record by 0.25°C Terhaar & Burger (2025). This change has significantly altered global ocean current systems Marcos et al. (2025), modifying marine vertical stability and transforming seawater temperature-density structures, thereby affecting the enthalpy flux process through which TCs extract energy from oceans. As one of Earth's most devastating natural disasters, TCs threaten coastal regions with destructive winds, extreme precipitation, and storm surge Wang et al. (2025). However, accurate prediction of TC trajectories and intensities remains challenging due to the triple influences from atmospheric circulation, oceanic, and land systems, in addition to their complex physical processes Kang et al. (2024).

With the continuous deepening of research on the physical mechanisms of TC motion, a growing body of findings has demonstrated that the environmental field exerts a significant impact on the track and intensity of TCs, which also reveals the complex characteristics of TC track and intensity changes Kang et al. (2024). Among these factors, the combined effects of ocean heat content, atmospheric dynamical processes, and underlying surface topography are particularly crucial. As shown in Figure 1, the interactions between these factors further amplify the complexity of TC prediction. However, due to the heterogeneity and multimodality of relevant data, the current state-of-the-art

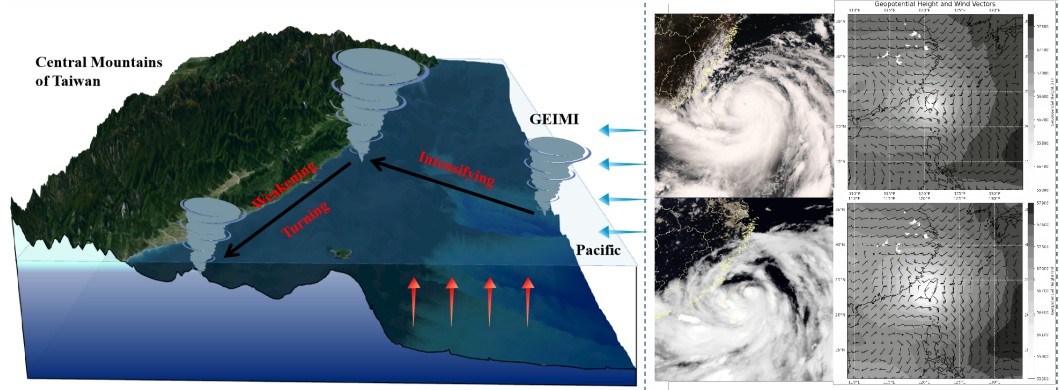

Figure 1: Abnormal deflection of TC GEIMI during landfall in Taiwan in 2024. The Central Mountain Range (peaking at nearly 4,000 m) interacted with GEIMI's low-level circulation within the 600–700 hPa vertical layer, inducing a Venturi effect that guided its low-level center southward. Satellite imagery showed the eye of GEIMI became less distinct following its interaction with the terrain.

foundational models for TC prediction fail to effectively integrate these key influencing factors, thereby neglecting the critical information contained in other physical driving factors that affect TC movement. Recent studies have laid the groundwork for the assessment of extreme TC disasters in the South China Sea buffer zone, but they have also highlighted the enormous challenges faced in short-term operational forecasting Ma et al. (2025). This is because current high-resolution regional models fail to accurately capture the forcing effects of these islands, while deep learning models completely ignore the influence of topography, including the uncharacterized complex interactions among atmospheric, oceanic, and topographic processes. This also reveals a gap in understanding the complex impact of topography on TC behavior. Similar blocking effects caused by islands play a key role in other cyclone-prone regions, such as the Taiwan Strait and the Mozambique Channel. These situations underscore the necessity of further advancing the atmospheric-oceanic-topographic coupled modeling framework. Addressing this gap is of vital significance for improving operational TC prediction, extending the forecast lead time, and ultimately enhancing disaster prevention, preparedness, and response strategies.

In addition, the generation of TCs in the Western North Pacific (WNP) is regulated by multiple factors such as seasonal atmospheric circulation and oceanic systems, exhibiting significant multimodal characteristics: the average origin location and trajectory curves are closely related to large-scale air-sea patterns. Existing studies have clearly identified several classical TC pattern classifications in this region Fu et al. (2025), while the latest research points out that these patterns are essentially multimodal distributions composed of multiple disconnected sub-patterns Zhu et al. (2025). Traditional TC forecasting models often face the problem of high computational cost. Moreover, due to the adoption of fixed distribution assumptions, they do not consider the discontinuity between patterns, making it difficult to learn such multimodal distributions from continuous latent spaces and easily leading to model collapse Hoang et al. (2018); Dendorfer et al. (2021). Another key limitation of existing methods lies in the issue of "catastrophic forgetting", that is, when new TC patterns emerge, the pre-trained parameters will drift towards the new distribution, resulting in a significant decline in the prediction performance of original TC sub-patterns. To address the above challenges, we propose a Mixture of Experts (MoE) architecture, where each expert module is adept at capturing specific patterns of tropical cyclones. This method introduces an input-adaptive gating mechanism, which can select and combine relevant expert modules according to the input physical environmental field, realizing efficient parameter sharing.

To address these challenges, we propose AOT-TCNet, a multimodal TC forecasting model that integrates data governing oceanic, terrestrial, and atmospheric processes to comprehensively characterize key physical factors influencing TC motion and intensity variations. This cross-domain data coupling approach is crucial for enhancing TC prediction accuracy. The main contributions of this work are as follows:

- We pioneer the incorporation of complex underlying terrain elevation under TC conditions into a multimodal deep learning framework, which enables explicit quantification of physical mechanisms such as terrain-lifted wind acceleration and anomalous north–south deflection, effectively addressing the forecasting challenges for abnormally deflected TCs.

- We present the most comprehensive and spatiotemporally extensive TC dataset to date for the WNP, encompassing 1,808 tropical cyclone tracks and associated intensity records from 1950 to 2024, with integrated multi-domain variables from the atmosphere, ocean, and land.

- We propose AOT-TCNet, the first deep learning model that unifies atmosphere, ocean, and terrain inputs under a coupled modeling framework. Incorporating the TC-Mode-Adaptive Mixture of Experts (TMA-MoE) architecture enhanced by the Reinforcement Learning with Physical Feedback (RLPF) raining strategy.

  Extensive experiments on all tropical cyclone events from 2017 to 2024 show that AOT-TCNet achieves state-of-the-art tropical cyclone forecasting performance, particularly excelling in predicting abnormally deflected tropical cyclones that were previously considered difficult to forecast. Some indicators even surpass the prediction level of official forecasting systems.

## 2 RELATED WORK

### 2.1 DEEP LEARNING-BASED TC FORECASTING

In recent years, tropical cyclone prediction remains one of the core challenges in the field of meteorology. Traditional prediction methods primarily rely on numerical weather prediction models, yet these models suffer from high computational resource consumption and limited real-time capability for short-term operational forecasting. Deep learning, leveraging its adaptive learning capability for complex spatiotemporal patterns and faster inference speed, has achieved remarkable progress in the field of weather forecasting Chen et al. (2023); Bi et al. (2023); Lam et al. (2023) and has become a research frontier in this domain. In TC prediction, early studies drew on the idea of trajectory prediction Alahi et al. (2016) and employed recurrent neural networks (RNNs) and their variants for path prediction Moradi Kordmahalleh et al. (2016); Alemany et al. (2019); Pan et al. (2019); Gao et al. (2018). With the deepening of research, multimodal fusion architectures have been extensively explored Giffard-Roisin et al. (2020); Wang et al. (2025). The integration of deep learning with meteorological data has not only improved prediction accuracy but also laid the foundation for constructing more reliable and efficient tropical cyclone forecasting models. Although these methods capture the interactions between the atmosphere and the ocean, they do not take into account the role of topographic constraints, including the buffering effect of continuous topographic blocking and the venturi effect induced by topographic lifting Ma et al. (2025). This highlights the necessity of further advancing the atmosphere-ocean-terrain coupled modeling framework to enhance the adaptability, robustness, and forecasting accuracy.

### 2.2 GENERATIVE MODEL-BASED TC FORECASTING

Affected by complex weather systems, TC paths exhibit high diversity. Under extreme meteorological conditions, it is even difficult to obtain a single credible trajectory, and challenges remain in the forecasting of TCs with abnormal deflection. Generative models have been extensively studied in trajectory prediction tasks Kosaraju et al. (2019); Amirian et al. (2019); Dendorfer et al. (2020); Fernando et al. (2018). Such methods aim to generate the possible distribution of future trajectory states. Inspired by these methods, Rüttgers et al. Rüttgers et al. (2019) proposed a tropical cyclone path prediction method based on generative adversarial networks, and verified the potential of deep learning in remote sensing data prediction through satellite images. Huang et al. Huang et al. (2022) proposed the multimodal trajectory prediction network MMSTN, which adopts a generative adversarial network architecture to generate multimodal future trajectories as feasible candidates. On this basis, they further proposed the Multi-source Meteorological Data Efficient Utilization Method Huang et al. (2023; 2025), confirming the effectiveness of deep learning in capturing complex atmospheric patterns. As powerful generative frameworks, generative adversarial networks (GAN) have advantages in modeling multimodal trajectory distributions. However, existing methods are still trained under supervised frameworks and rely on a single real trajectory Subich et al. (2025), making it difficult to overcome mode collapse Gupta et al. (2018); Sadeghian et al. (2019); Goodfel-

low et al. (2014). MoE systems and reinforcement learning have been proven effective in enhancing reasoning capabilities. They can guide the behavior of multi-expert networks based on multi-level reward signals, leading the expert networks to optimize realism, diversity, and physical consistency simultaneously.

## 3 METHOD

TC forecasting aims to predict future trajectories and intensity of TCs given historical states. Specifically, we take the time series $X = \{x_{\text{lon}}, x_{\text{lat}}, x_{\text{traj}}, x_{\text{int}}\}$ consisting of longitude, latitude, central pressure, and wind speed as input. For a given historical sequence $\{X_{-t+1}, X_{-t+2}, ...X_0\}$ with fixed temporal length, the objective is to predict the future trajectory and intensity , where each trajectory contains the subsequent $\hat{Y} = \{\hat{Y}_{\text{lon}}, \hat{Y}_{\text{lat}}, \hat{Y}_{\text{traj}}, \hat{Y}_{\text{int}}\}$ time steps of positional and intensity evolution.

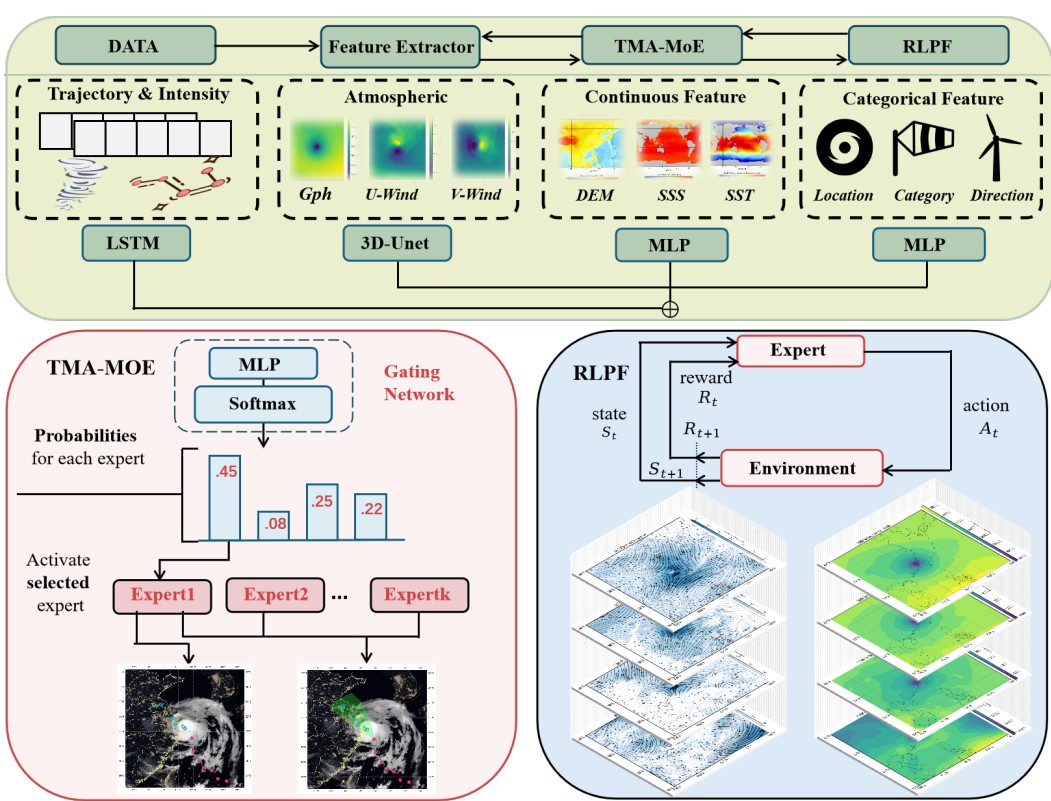

Figure 2: The architecture of AOT-TCNet, comprising Encoders, TMA-MoE, and RLPF. The environment data $D$ and observed trajectories $X$ are encoded and passed to the fusion module. The experts can predict different mode trajectory distributions for the given observation. The gating network estimates probabilities $\pi$ for the experts. The model samples or selects a expert from $\pi$ and predicts a trajectory $Y$ conditioned on the features $c$ and the noise vector $z$.

### 3.1 ENCODERS

We outline the architecture of our model in Figure 2. First, we adopt a layered embedding strategy to independently encode the TC, ocean, terrain, and atmospheric data, aiming to capture the co-evolutionary characteristics of variables across different physical domains during TC development. After encoding, the features are concatenated into c via a cross-modal feature alignment mechanism. The concatenated features are then fed into a two-layer MLP, enabling the model to simultaneously learn and capture correlations across the ocean, terrestrial, and atmospheric domains. This module design allows the model to comprehensively consider key factors such as atmospheric circulation,

ocean heat content, and underlying surface variations, thereby improving the accuracy and generalization capability of TC prediction. The details of the encoders are in Appendix A.2.

## 3.2 TC-Mode-Adaptive Mixture of Experts

The MoE model enhances the performance of Large Language Models (LLMs) by integrating the advantages of multiple sub-models Jacobs et al. (1991); Shazeer et al. (2017); Chen et al. (2022); Zhong et al. (2022). Recently, Chowdhury Chowdhury et al. (2023) theoretically pointed out that the MoE gating network can directly direct input embeddings with similar class patterns to the same expert while filtering out class-irrelevant features. This enables the gating network to selectively activate a subset of experts for each mode embedding. Compared with activating all experts simultaneously, it can achieve feature learning for specific modes while reducing computational overhead. This characteristic provides new ideas for TC forecasting in the meteorological field. Therefore, we introduce the MoE module into the TC forecasting task. This module uses the MoE gating network to activate experts for specific patterns, treating the multi-modal target distribution as a mixture of multiple continuous trajectory distributions to learn different categories of TC patterns.

**Mode-Adaptive-Expert.** We reconstruct multiple independent generative networks into an MoE structure (rather than rewriting standard linear or convolutional layers). The core purpose is to enable each expert sub-model to specially learn one type of TC track pattern distribution, approximate the real data distribution through the mixture of multi-modal distributions, and at the same time prompt each expert to focus on different TC patterns. These expert sub-models adopt the same network structure but do not share weights, and selectively activate specific experts based on the encoded feature $c$ of TCs. Each generative network containing an activation function is regarded as an expert, initialized with the encoded feature $c$, and random noise $z$ is added as the initial hidden state. The calculation process of the predicted trajectory is as follows:

$$h_i = c_i + z_i \tag{1}$$

$$\hat{Y}_i = Expert\,(X_i, h_i) \tag{2}$$

where $e$ denotes the expert index and $z_j$ represents the random noise vector. We decompose the expert group into $E = (E_1, E_2, ..., E_k)$, and at this time, the output can be written as:

$$Y_i = [E_1(X_i, h_i), E_2(X_i, h_i), ..., E_k(X_i, h_i)] \tag{3}$$

where $E_k(\cdot)$ denotes the generating function of the $k_{th}$ expert.

**Gating Network.** The gating network is an independent and learnable sub-network, and its routing strategy is part of the end-to-end process. The gating network takes the encoded features as input and outputs the correlation probabilities between the encoded features and all experts, thereby activating the most relevant experts. This network consists of a 3-layer MLP with ReLU activation and a hidden dimension of 48. To estimate the probability of experts, we use the distribution of expert-predicted trajectories to describe the likelihood of real TC trajectories:

$$p(Y_i \mid c_i, E_k) \propto \exp\left(-\frac{|E_k(X_i, h_i) - Y_i|_2^2}{2\sigma^2}\right) \tag{4}$$

Then, through Bayes' rule, we can obtain the posterior distribution of experts:

$$p(E_k \mid c_i, Y_i) = \frac{p(Y_i \mid c_i, E_k)}{\sum_{j=1}^{K} p(Y_i \mid c_i, E_j)} \tag{5}$$

Finally, we optimize the gating network using the cross-entropy loss between the conditional probability of each expert and the output distribution of the gating network:

$$\mathcal{L}_\Pi = \frac{1}{n} \sum_{i=1}^{n} H\left(p(E \mid c_i, Y_i), \pi(c_i)\right) \tag{6}$$

Where $|| \cdot ||_2^2$ denotes the squared $L_2$ norm, $\sigma$ is the Gaussian kernel bandwidth parameter, and $\pi(c_i) = [\pi_1(c_i), \pi_2(c_i), ..., \pi_k(c_i)]$ is the expert probability output by the gating network. Ultimately, the network is trained to route to the expert that generates the trajectory pattern most consistent with the current TC path.

### 3.3 REINFORCEMENT LEARNING WITH PHYSICAL FEEDBACK

Reinforcement Learning from Human Feedback (RLHF) has attempted to train reward models by collecting human feedback on generation tasks Christiano et al. (2017); Stiennon et al. (2020); Ouyang et al. (2022), thereby optimizing the output performance of models. Drawing on this idea, we propose an RLPF framework: TC forecasting is modeled as a Markov Decision Process (MDP) with a physics-consistent reward mechanism. To achieve this, we use the inherent consistency of physical laws is used to replace human feedback, and a reward signal is constructed by quantifying the degree of agreement between the TC trajectory and the physical constraints of the atmospheric environment (i.e., "physical feedback"). This signal guides the model to generate trajectories that are not only accurate but also consistent with physical mechanisms. To further constrain the rationality of outputs, we have learned a reward function $r_t(x, c)$ based on physics-consistency feedback:
$State : s_i = (\hat{Y}_i^{\alpha_i}, c_i), Action : a_i = \hat{Y}_i^t, Reward : r_i = r_{t,i} + r_{phy,i} + r_{\text{mode},i}.$

Formally, the reward function is trained by maximizing $r(t, i)$:

$$r_{t,i} = - \begin{cases} \frac{1}{2} \left( Y_i^t - \hat{Y}_i^t \right)^2, & \text{if } \left| Y_i^t - \hat{Y}_i^t \right| < \delta, \\ \delta \left| Y_i^t - \hat{Y}_i^t \right| - \frac{1}{2}\delta^2, & \text{otherwise.} \end{cases} \tag{7}$$

Where $\delta$ denotes the threshold of the similarity reward.

The physical consistency of the reward function $r_{phy}$ is trained by maximizing $r(phy, i)$:

$$r_{phy,i} = -\frac{1}{n} \sum_{i=1}^{n} \left| Y_i^{atm} - \hat{Y}_i^{atm} \right| \tag{8}$$

Furthermore, we introduce a diversity reward $r(mode, i)$ to incentivize experts to cover different TC track patterns. The diversity reward between the classifier's output and the true expert labels of the predicted trajectories urges experts to simulate non-overlapping distributions and ensures that the trajectories of different experts are spatially separated:

$$r_{\text{mode}} = \sum_{E=1}^{k} \log C_E \left( \hat{Y}_i \right) \tag{9}$$

Where $k$ represents the number of experts.

In summary, RLPF employs a reward mechanism to iteratively train the TMA-MoE system. The forecasting model is formulated as an MDP with physics-consistent constraints, and the reward is used to update the RL policy, thereby concluding one phase of the loop. RL updates in accordance with the following policy:

$$J(\theta) = \mathrm{E}_{\substack{(c,Y \sim D_{atm}) \\ \hat{Y} \sim \pi_\theta}} \left[ \sum_{t=0}^{T_{pred}} \gamma^t r_t \right] \tag{10}$$

Where $J(\theta)$ denotes the objective function of the RL policy, $\theta$ represents all learnable parameters of the model, $D_{atm}$ stands for the physical environment dataset, $\pi(\theta)$ is the predicted distribution of generated trajectories with parameter $\theta$, $\gamma$ is the discount factor, and E denotes the expectation operator.

## 4 EXPERIMENTS

### 4.1 EXPERIMENTAL SETTINGS

**AOT-TCs Dataset.** We collected atmospheric, terrain, and oceanic datasets associated with TCs from 1950 to 2024, closely related to TC genesis and decay processes. The data sources include the *CMA-BST Dataset*, *ERA5 Reanalysis Data*, *CODCv1*, and *GEBCO*. By integrating these heterogeneous data sources, the *AOT-TCs* dataset enriches the information dimensions for TC prediction models, thereby improving forecast accuracy and advancing the understanding of complex physical mechanisms. The details of the dataset are in Appendix A.1.

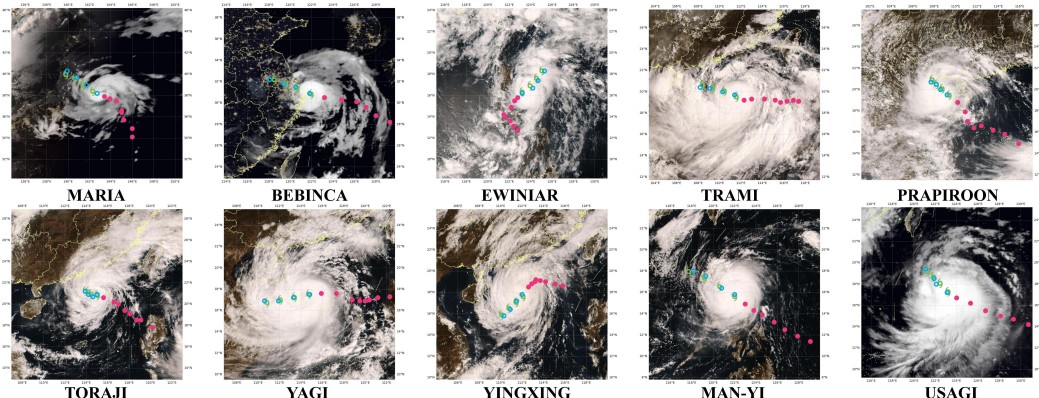

Figure 3: Forecast results for selected TCs in 2024. Historical trajectories are shown in red, ground truth future paths in blue, and model-predicted trajectories in green.

**Implementation Details.** We conduct experiments using an NVIDIA GTX 4090D GPU. The model is optimized with the Adam optimizer, configured with an initial learning rate of 0.0001, a batch size of 128, and trained for 300 epochs. In our experiments, we input historical TC trajectory data from the past 48 hours (8 timesteps) to predict future TC trends for the next 24 hours (4 timesteps).

**Evaluation Metrics.** We adopt the same evaluation configuration as before Huang et al. (2025). For path forecasting, we calculate the absolute distance between the actual position and the forecast position, and evaluate the average position error between the predicted trajectory and the GT trajectory. For intensity forecasting, we calculate the mean absolute error between the actual air pressure (hPa) and wind speed (m/s) and the forecasted air pressure and wind speed.

### 4.2 COMPARISON WITH STATE-OF-THE-ARTS

On the 2017–2019 test set, our method was comprehensively compared against various existing techniques, including pedestrian trajectory prediction models, mainstream tropical cyclone forecasting methods, and official forecasts from the China Meteorological Observatory (CMO) C.M.O (2019). As shown in Table 1, our approach outperforms the compared models in most cases, particularly excelling in ultra-short-term forecasting: the track prediction errors at 6, 12, and 18-hour lead times were reduced by 13 km, 10.95 km, and 1.11 km, respectively, compared to CMO. However, CMO performs better in 24-hour forecasting, and as the lead time increases, numerical weather prediction (NWP) methods gradually demonstrate stronger performance.

In terms of intensity forecasting, our model improves accuracy by 53% to 71.8% over CMO, and this advantage remains consistent across all forecast lead times. Compared to other deep learning models, our method also exhibits competitive performance: although the 6-hour track error is slightly higher than that of the recent $TCN_m$ Huang et al. (2025), it achieves state-of-the-art results at longer forecast horizons.

It is worth noting that many advanced deep learning models that perform well in pedestrian/vehicle trajectory prediction show limited effectiveness in tropical cyclone forecasting, which stems from the fundamental differences between the two tasks: the former focuses on local interactions and road conditions, while TC forecasting is governed by complex physical processes. Our model can effectively handle larger and more diverse datasets. The results indicate that forecast accuracy is strongly correlated with the availability of relevant meteorological data because richer data enables the model to extract more discriminative features, thereby improving predictive performance. Finally, Figure 3 presents case studies of TC forecasts from 2024, further validating the superior performance of our proposed model.

### 4.3 COMPARISON WITH GLOBAL OPERATIONAL FORECASTING METHODS

In addition to deep learning-based approaches, we further compared our model against NWP systems, including forecasts from official meteorological agencies across multiple countries. As sum-

Table 1: Comparisons of average absolute error of TC prediction of different methods

| Methods | Dist (km) ↓ | | | | Pres (hpa) ↓ | | | | Wind (m/s) ↓ | | | |
|---|---|---|---|---|---|---|---|---|---|---|---|---|
| | 6h | 12h | 18h | 24h | 6h | 12h | 18h | 24h | 6h | 12h | 18h | 24h |
| SGAN | 28.88 | 61.75 | 98.74 | 140.61 | 1.91 | 3.12 | 4.2 | 5.12 | 1.05 | 1.69 | 2.28 | 2.81 |
| GBRNN | 29.93 | 65.06 | 105.74 | 152.06 | - | - | - | - | 1.16 | 1.89 | 2.52 | 3.1 |
| MMSTN | 27.57 | 59.09 | 96.54 | 139.19 | 1.69 | 2.86 | 3.94 | 4.74 | 0.95 | 1.52 | 2.1 | 2.55 |
| MGTCF | 23.14 | 43.37 | 67.09 | 93.08 | 1.37 | 2.04 | 2.66 | 3.29 | 0.73 | 1.17 | 1.55 | 1.86 |
| $TCN_m$ | **22.98** | 43.83 | 66.41 | 93.76 | - | - | - | - | 0.7 | 1.09 | 1.43 | 1.75 |
| CMO | 37.08 | 52.93 | 60.69 | **75.49** | 2.67 | 4.3 | 5.04 | 6.31 | 2.29 | 3.45 | 2.75 | 5 |
| Our | 24.03 | **41.98** | **59.58** | 86.05 | **1.22** | **1.74** | **2.19** | **2.58** | **0.71** | **1.03** | **1.23** | **1.41** |

marized in Table 2, our method achieved state-of-the-art performance in intensity forecasting and demonstrated competitive accuracy in 12-hour trajectory predictions. While our trajectory predictions did not surpass the best-performing NWP model, our approach outperformed other NWP models using only a single GPU, while improving computational efficiency by several orders of magnitude. This result highlights the rapid-response capability of our model in ultra-short-term forecasting scenarios and underscores its robustness in resource-constrained environments. The ability to deliver accurate predictions with limited computational resources indicates the model's scalability and practical applicability for real-world deployment.

Table 2: Comparison with global operational models, including the China Meteorological Administration (CMA), Joint Typhoon Warning Center (JTWC), Korea Meteorological Administration (KMA), Japan Meteorological Agency (JMA), and Hong Kong Observatory (HKO).

| Methods | 2020 | | | | 2021 | | | | 2022 | | | |
|---|---|---|---|---|---|---|---|---|---|---|---|---|
| | Dist (km) ↓ | | Wind (m/s) ↓ | | Dist (km) ↓ | | Wind (m/s) ↓ | | Dist (km) ↓ | | Wind (m/s) ↓ | |
| | 12h | 24h | 12h | 24h | 12h | 24h | 12h | 24h | 12h | 24h | 12h | 24h |
| CMA | - | 73.5 | - | 4.5 | 56.7 | 88.3 | - | 4.4 | 51.9 | 76.5 | - | 4.5 |
| JMA | - | 74.2 | - | 4.5 | 51.3 | 85.3 | - | 4.4 | **48.4** | **72.7** | - | 5.5 |
| JTWC | - | 74.9 | - | 4.9 | 56.6 | 89.3 | - | 4.8 | 55.6 | 78.2 | - | 5.4 |
| KMA | - | 88.3 | - | 4.7 | 61 | 97.9 | - | 4.9 | 59.8 | 83.8 | - | 5.1 |
| HKO | - | **70.6** | - | 4.7 | - | **83.2** | - | 4.9 | - | 79.3 | - | 5.2 |
| Our | **42.04** | 80.86 | **1.28** | **1.72** | **49.57** | 88.3 | **1.17** | **1.62** | 48.94 | 104.72 | **1.26** | **1.56** |

## 4.4 ABLATION STUDY

To validate the effectiveness of the proposed method, we conducted extensive ablation studies. As shown in Table 3, the experiments compare the following models: a baseline trained only on the *CMA-BST* dataset, a model trained on *ERA5* data, a model trained on both *ERA5* and *AOT-TCs* data, and our full model incorporating the TMA-MoE system. The results demonstrate that each component contributes positively to performance, with the integrated model achieving the best outcome. Notably, the incorporation of heterogeneous data significantly improves both track and intensity forecasting, indicating the effectiveness of the proposed data sources and encoder architecture for TC prediction, and highlighting strong correlations between TC characteristics and the integrated geophysical variables. Furthermore, the MoE framework enhances prediction stability and mitigates the over-smoothing issue observed in previous models, underscoring the robustness and reliability of our framework.

## 4.5 ENSEMBLE FORECASTING OF ABNORMALLY DEFLECTED TCS

Abnormally deflected TCs are characterized by limited sample availability and rapid path shifts, posing significant challenges for prediction. Existing forecasting models often struggle with these cases due to over-smoothed outputs and the absence of physically grounded constraints, making it difficult to capture such sharp turning behaviors. To ad-

Table 3: Comparative Results from the Ablation Study

| Components | | | Dist (km) ↓ | | | | Pres (hpa) ↓ | | | | Wind (m/s) ↓ | | | |
|---|---|---|---|---|---|---|---|---|---|---|---|---|---|---|
| E | A | M | 6h | 12h | 18h | 24h | 6h | 12h | 18h | 24h | 6h | 12h | 18h | 24h |
| | | | 27.99 | 60.76 | 97.85 | 141.73 | 1.94 | 3.23 | 4.34 | 5.18 | 0.95 | 1.58 | 2.11 | 2.58 |
| ✓ | | | 26.66 | 50.77 | 76.49 | 108.72 | 1.25 | 2.12 | 2.86 | 3.53 | 0.74 | 1.15 | 1.53 | 1.85 |
| ✓ | ✓ | | 25.31 | 48.01 | 72.25 | 101.01 | 1.27 | 1.94 | 2.62 | 3.17 | 0.70 | 1.13 | 1.47 | 1.76 |
| ✓ | ✓ | ✓ | **24.03** | **41.98** | **59.58** | **86.05** | **1.22** | **1.74** | **2.19** | **2.58** | **0.71** | **1.03** | **1.23** | **1.41** |

Table 4: Comparison of different methods for abnormally deflected TC prediction

| Method | Dist (km) ↓ | | | | Pres (hpa) ↓ | | | | Wind (m/s) ↓ | | | |
|---|---|---|---|---|---|---|---|---|---|---|---|---|
| | 6h | 12h | 18h | 24h | 6h | 12h | 18h | 24h | 6h | 12h | 18h | 24h |
| MMSTN | 38.47 | 73.61 | 94.82 | 173.72 | 2.05 | 3.04 | 4.01 | 5.26 | 1.08 | 1.52 | 2.14 | 2.64 |
| MGTCF | **25.21** | 49.72 | 71.04 | 107.80 | 1.57 | 2.27 | 3.23 | 4.05 | 0.92 | 1.23 | 1.82 | 2.06 |
| Our | 26.16 | **47.39** | **64.76** | **93.29** | **1.54** | **2.02** | **2.65** | **3.04** | **0.83** | **1.02** | **1.31** | **1.56** |

dress this, we leveraged a MoE framework combined with surface elevation information to enhance the model's capability for abnormal scenario simulation and causal reasoning. Our approach successfully predicted the behavior of abnormally deflected TCs. We identified and reported all such cases in the test set individually, and the results, presented in Table 4, show that our method consistently achieved the best performance. These findings underscore the critical role of physical variables and generative modeling in predicting anomalous TC deflections. Furthermore, we conducted a detailed case analysis of the abnormally deflected TCs in 2024, accompanied by ensemble forecast visualizations. As illustrated in Figure 4, AOT-TCNet effectively captured both the sudden deflection and the overall deflection trend. Although some positional errors remain, the results clearly demonstrate the model's robust capability for ensemble forecasting of abnormally deflected TCs.

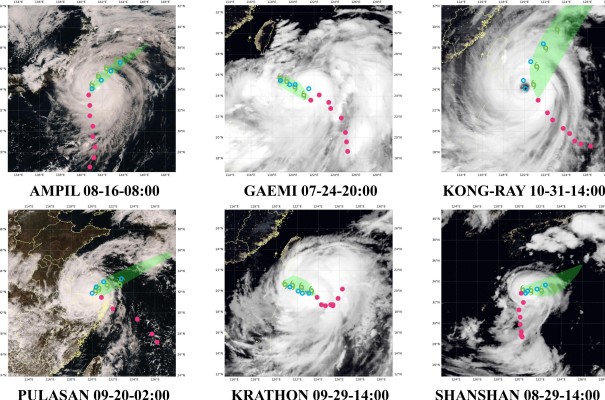

**AMPIL 08-16-08:00**    **GAEMI 07-24-20:00**    **KONG-RAY 10-31-14:00**

**PULASAN 09-20-02:00**    **KRATHON 09-29-14:00**    **SHANSHAN 08-29-14:00**

Figure 4: Ensemble forecasting results for abnormally deflected TCs in 2024. Historical trajectories are shown in red, actual future tracks in blue, predicted paths in green, and shaded regions denote the projected trajectory trends.

## 5 CONCLUSION

This study proposes the first multimodal TC prediction framework based on atmosphere-ocean-terrain coupled modeling, which improves the prediction accuracy of TC tracks and intensities by fully coupling heterogeneous geophysical variables. In addition, we present a RLPF-based TMA-MoE architecture that enhances forecasting stability through multi-expert forecasting and physical consistency constraints. Comparisons with state-of-the-art deep learning models and official forecasts show that this method offers significant advantages in terms of both accuracy and cost-effectiveness, with notable performance improvements in short-term predictions and complex scenarios involving anomalously deflected TC tracks.

THICS STATEMENT

In compliance with the ICLR Code of Ethics, all authors of this paper have read and adhered to the principles outlined therein. This study does not involve human subjects or personal data collection. All datasets used in this work are publicly available and have been properly cited.

REPRODUCIBILITY STATEMENT

To promote academic exchange and ensure the reproducibility of this research, the experimental code and data involved in this thesis have been made publicly accessible:

- **Datasets and Code Repository:** `https://anonymous.4open.science/r/AOT-TC-111F`

We affirm that the provided code can reproduce the main experimental results reported in this paper.

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

## A Appendix

### A.1 Dataset

We collected atmospheric, terrain, and oceanic datasets related to TCs from 1950 to 2024, which are closely associated with TC genesis and decay. This comprehensive dataset allows for a holistic exploration of TCs, enabling the revelation of their evolutionary patterns and characteristics throughout their lifecycle.

**CMA-BST Dataset.** TC track and intensity characteristics were obtained from the *China Meteorological Administration Tropical Cyclone Best Track Dataset(CMA-BST)*, which records key information for each TC lifecycle in the Western North Pacific and South China Sea, including longitude, latitude, minimum central pressure, and 2-minute average maximum near-center wind speed.

**ERA5 Reanalysis Data.** The *ERA5 reanalysis dataset*, provided by the European Centre for Medium-Range Weather Forecasts (ECMWF), offers a spatial resolution of 0.25°×0.25° with 6-hourly updates. We utilized wind fields and geopotential height to characterize TC atmospheric circulation and pressure structures. Specifically, centered on the current cyclone position, we extracted the u-wind, v-wind, and geopotential height within a 25°×25° spatial window around the TC center. Inspired by statistical prediction models and sensitivity analyses, and to capture air movements in the lower, middle, and upper troposphere, pressure levels at 200, 500, 850, and 1000 hPa were selected.

**Global Ocean Science Database (CODCv1).** The CODCv1 in-situ ocean observation dataset, provided by the CAS-Ocean Data Center, was employed to analyze TC intensity, which is regulated by energy exchange between the ocean and atmosphere—particularly surface evaporation flux. Recent studies have demonstrated that the upper-ocean salinity barrier layer (BL) induces specific upper-ocean responses to Western North Pacific TCs: during TC passage, surface salinity significantly (SSS) increases on both sides of the track, while SST slightly decreases. The BL limits TC-induced SST cooling by suppressing cold water entrainment from the thermocline. Therefore, SST and salinity were selected as predictors for TC intensity changes.

**GEBCO Dataset.** The Global Bathymetric Dataset from the Intergovernmental Oceanographic Commission (IOC) and International Hydrographic Organization (IHO) provides high-precision global seafloor topography data from coastlines to deep oceans, including detailed representations of mid-ocean ridges, trenches, and plateaus.

In addition to conventional gridded data, we incorporated critical non-spatial variables to enhance the dataset's multidimensional information. These variables include the translational speed of TC centers, historical movement direction, historical intensity variability, month, and the Niño3.4 index. Historical movement direction and intensity variability provide essential prior information for TC trajectory and intensity forecasting; monthly data reflect seasonal cyclicity characteristics, with significant differences in formation and movement patterns of western Pacific TCs across seasons; translational speed is intrinsically linked to TC lifecycle dynamics. Furthermore, recent studies demonstrate that average TC genesis locations and trajectory modes exhibit significant ENSO (El Niño-Southern Oscillation) covariability Fu et al. (2025), with characteristic oceanic thermal response patterns observed during TC development stages. By integrating these heterogeneous data sources, the *AOT-TCs* dataset enriches TC prediction models with enhanced informational dimensionality, thereby improving forecast accuracy and advancing understanding of complex physical processes.

In summary, we present *AOT-TCs*, the longest and most comprehensive TC dataset for the Northwest Pacific, covering the period from 1950 to 2024. Unlike existing public datasets, *AOT-TCs* integrates multimodal information across diverse physical variables influencing TC dynamics rather than relying on single-sensor observations, thereby establishing a unified framework for TC prediction research. We denote *AOT-TCs* as $D_{x,c} = \{(x_i, c_i)\}_{i=1}^{n}$, which consists of $n$ TC data sequences $x$ and corresponding environmental variable data $c$ sampled from a joint distribution $P(x, c)$. Our goal is to develop $k$ experts $E_k$ such that the generated predicted TC sequence $E_i(x, c)$ matches $P(x|c)$ in distribution.

All data used in the analysis are available in public repositories.

1) ERA5 data can be download from `https://cds.climate.copernicus.eu/cdsapp#!/dataset/reanalysis-era5-single-levels`

2) CMA-BST data can be download from `https://tcdata.typhoon.org.cn/zjljsjj.html`

3) GEBCO data can be download from `https://www.gebco.net/data-products/gridded-bathymetry-data`

4) CODCv1 data can be download from `https://www.casodc.com/data/`

## A.2 The Implementation Details of Encoders

Consider a multivariate TC sequence sample $X$, where each sample $X$ is associated with environment data $D$ comprising categorical data $D_{cat} \in \mathbb{N}_{L \times K_{cat}}$, continuous data $D_{cont} \in \mathbb{N}_{L \times K_{cont}}$, and atmospheric circulation data $D_{atm} \in \mathbb{N}_{L \times K_{atm}}$. We employ a layered embedding strategy to independently encode TC, ocean, terrain, and atmospheric data. Through a cross-modal feature alignment mechanism, we perform structured information encoding to capture the co-evolutionary characteristics of variables across different physical domains during TC development. This ensures that data from diverse sources are appropriately mapped into a unified feature space.

**TC-Encoder.** For TC sequence samples, we utilize LSTM to encode historical trajectories and extract dynamic features. The TC $X_i$ are encoded into high-dimensional features $\theta_{TC}$ through the following formulation:

$$c_{TC} = \text{LSTM}_{\text{Encoder}}(X_i^t, h_i^t) \tag{11}$$

Where $h_i^t$ denotes the hidden state of the LSTM at time step $t$.

**AOT-Encoder.** To efficiently integrate multimodal data from different physical domains into TC forecasting models and deeply explore their distinct physical processes and interaction mechanisms, we perform multi-scale feature extraction and fusion on ocean, terrestrial, and atmospheric data to construct a unified multimodal representation that is precisely embedded into existing TC prediction model architectures. First, each category in $D_{cat}$ is converted to one-hot encoding and then processed through a two-layer MLP to create categorical embedding $c_{cat}$. Similarly, $D_{cont}$ is encoded into continuous embedding $c_{cont}$ using a two-layer MLP. $D_{atm}$ employs CNN networks to extract spatiotemporal features from historical atmospheric circulation data, encoding them into atmospheric embedding $c_{atm}$:

$$c_{atm} = \text{3D-UNet}(D_{atm}(z_i, u_i, v_i)) \tag{12}$$

where $z$, $u$, $v$ represent geopotential height, meridional wind, and zonal wind respectively. Finally, these features are concatenated into $c_{condn} = c_{cat} + c_{cont} + c_{atm}$, where "+" denotes vector concatenation operation. The concatenated features are input through FC, enabling the model to simultaneously learn and capture correlations across ocean, terrestrial, and atmospheric domains. This module design allows the model to comprehensively consider atmospheric circulation, ocean heat content, underlying surface variations, and other key factors, thereby improving the physical consistency and generalization capability of TC predictions.

## A.3 Definition and Explanation of Abnormally deflected TCs

Abnormally deflected TCs refer to those that exhibit abrupt trajectory changes within a short time span due to the combined influence of multiple meteorological and environmental factors, including blocking highs, westerly troughs, the terrain of Taiwan Island, and binary TC interactions. In this study, we define an abnormal deflection event as a rightward turning angle exceeding 45° or a leftward angle exceeding 30° within a 12-hour period.

## A.4 Baselines

The compared methods are categorized into three main types: deep learning-based, generative model-based, and official operational numerical forecasting methods. We evaluate the proposed approach against the following baseline methods:

- Social gan Gupta et al. (2018): Socially acceptable trajectories with generative adversarial networks.

- GBRNN Alemany et al. (2019): Predicting hurricane trajectories using a recurrent neural network.

- MMSTN Huang et al. (2022): A multi-modal spatial-temporal network for tropical cyclone short-term prediction.

- MGTCF Huang et al. (2023): Multi-Generator Tropical Cyclone Forecasting with Heterogeneous Meteorological Data.

- $TCN_m$ Huang et al. (2025): Benchmark dataset and deep learning method for global tropical cyclone forecasting.

- C.M.O. C.M.O (2019): Typhoon network of central meteorological observatory. `http://typhoon.nmc.cn/web.html`.

### A.5 TC PATTERN CLASSIFICATIONS

Previous studies have identified several classical TC pattern classifications over the WNP Fu et al. (2025). To further investigate the spatial distribution characteristics of TCs under different trajectory patterns, we applied a K-means variant based on Dynamic Time Warping (DTW) to classify all TCs from 1950 to 2024 into seven distinct path clusters, as illustrated in Figure 5. These seven patterns effectively distinguish the typical TC trajectories across the basin. Clusters 1, 2, 4, and 7 are characterized by recurving paths. Cluster 2 consists mainly of TCs that form at relatively higher latitudes and follow a northeastward trajectory, typically representing oceanic recurving types. Clusters 1 and 7 are primarily generated over the eastern Philippine Sea; however, Cluster 1 tends to form farther west, resulting in a higher landfall rate and shorter lifespan. Clusters 3 and 6 both exhibit northwestward-moving trajectories, but differ in spatial extent—TCs in Cluster 3 originate east of the Philippines and move across the islands into the South China Sea, whereas those in Cluster 6 mostly form in the South China Sea and western Philippines, exhibiting shorter paths. Cluster 5 mainly includes TCs originating from the Northeastern Pacific, displaying a relatively unique cross-basin trajectory feature.

Table 5: Comparison of different methods for TC prediction from 2020 to 2024

| Methods | Dist (km) | | | | Pres (hpa) | | | | Wind (m/s) | | | |
|---|---|---|---|---|---|---|---|---|---|---|---|---|
| | 6h | 12h | 18h | 24h | 6h | 12h | 18h | 24h | 6h | 12h | 18h | 24h |
| MMSTN | 28.47 | 59.53 | 94.82 | 135.2 | 2.05 | 3.58 | 5.01 | 6.14 | 1.08 | 1.83 | 2.54 | 3.12 |
| MGTCF | **25.21** | _46.63_ | _71.04_ | _98.74_ | _1.57_ | _2.51_ | _3.23_ | _3.9_ | **0.83** | _1.39_ | _1.82_ | _2.25_ |
| Our | _27.16_ | **45.97** | **64.76** | **87.36** | **1.54** | **2.25** | **2.65** | **3.01** | _0.85_ | **1.23** | **1.41** | **1.65** |

### A.6 DETAILS ON TC PREDICTION EXPERIMENTS FROM 2020 TO 2024

Building upon our initial evaluation, we extended the TC forecasting experiments to include data from 2020 to 2024 and conducted additional comparisons with MMSTN and MGTCF. As shown in Table 5, our method continued to demonstrate strong performance, consistent with the results observed in the 2017–2019 period. We also observed that extreme values and intensities, which are becoming more frequent under global warming, tend to be underestimated when using limited training datasets. As climate change intensifies, it becomes increasingly critical to continuously update models with the most recent data to ensure accurate predictions.

### A.7 CASE ANALYSIS

We focus on analyzing our model's prediction of super TC YAGI. YAGI formed over the Northwestern Pacific east of the Philippines on September 1, 2024, made landfall in northeastern Philippines on September 2, and later entered the South China Sea. It made another landfall along the coast of Hainan from the afternoon to the night on September 6. During its trajectory, the TC reached a maximum sustained wind intensity of Category 17, resulted in 16 fatalities, and caused nearly 80 billion CNY in direct economic losses.

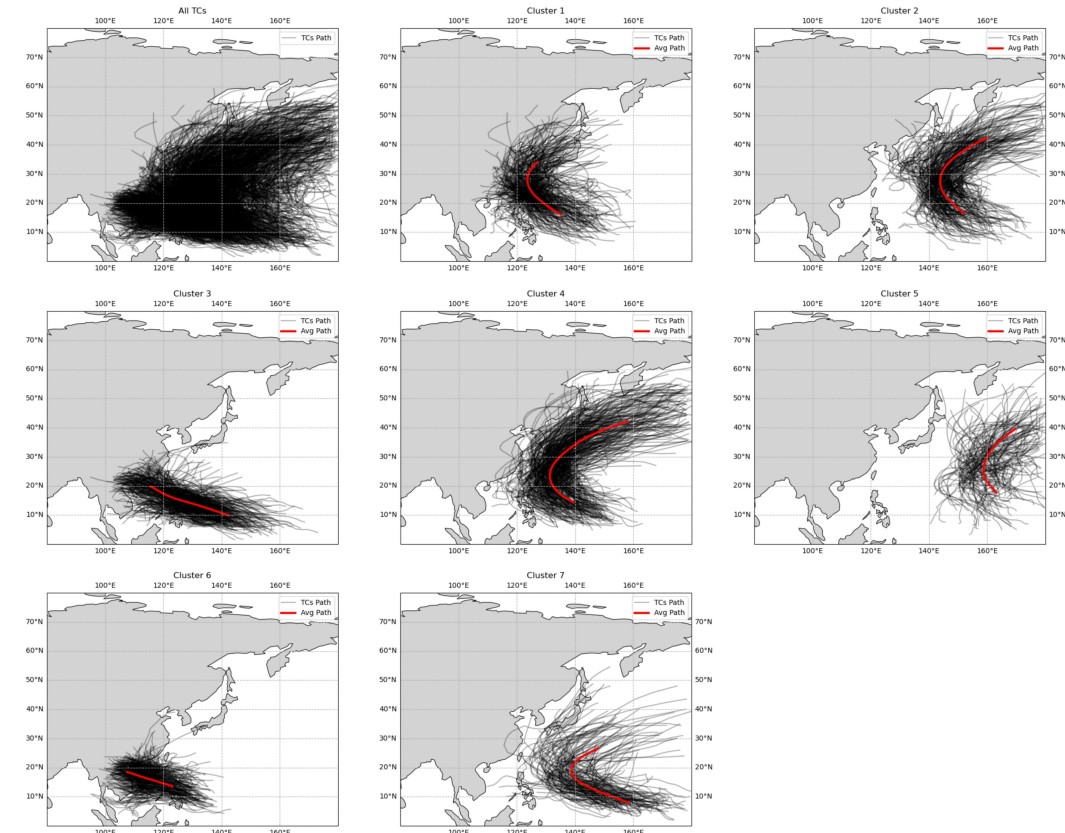

Figure 5: TC classification of different path patterns.

In the 00Z forecast on August 30, 2024, NCEP-GFS systematically underestimated the intensity of the subtropical high. Due to the high sensitivity of numerical models to initial conditions, even subtle errors can lead to significant deviations in long-term forecasts, which also explains the northward bias in the numerical prediction of TC tracks. NCEP-GFS predicted that YAGI would move north initially, then turn westward and make landfall in northern Fujian. In contrast, ECMWF-IFS HRES forecasted a stronger subtropical high, steering YAGI northwest toward Taiwan before entering Fujian. Our model demonstrated superior performance in predicting both the track and intensity of YAGI, successfully capturing its westward movement, as shown in Figure 6. In comparison, conventional numerical models exhibited a consistent northward bias in their track forecasts.

## A.8 LIMITATIONS & DISCUSSIONS & FEATURE WORK

Currently, both artificial intelligence models and traditional numerical models face significant challenges in accurately predicting TC anomalous paths caused by mesoscale circulation changes. Taking TC GEIMI in 2024 as an example, its looping motion before landfall was not accurately predicted in advance by any model Figure 8 shows our forecast results. Historical observation data indicate that multiple TCs have exhibited similar looping paths before making landfall in Taiwan, reflecting a certain universality of this phenomenon. The mechanism behind this looping motion is closely related to the northerly jet forced by Taiwan's topography. When a TC approaches the eastern coast of Taiwan, the Central Mountain Range blocks and accelerates the low-level inflow through a channeling effect, forming a significant northerly jet. This jet further guides the low-level center of the TC southward. Above the 600 hPa level, the topographic influence weakens significantly, while the upper-level circulation remains dominated by large-scale background flow, leading to vertical decoupling between the upper- and lower-level centers of the TC. This mechanism often causes the upper-level center to arrive over land first, creating an "illusion of landfall," whereas the actual trajectory of the low-level circulation, as shown in radar imagery, completes the looping

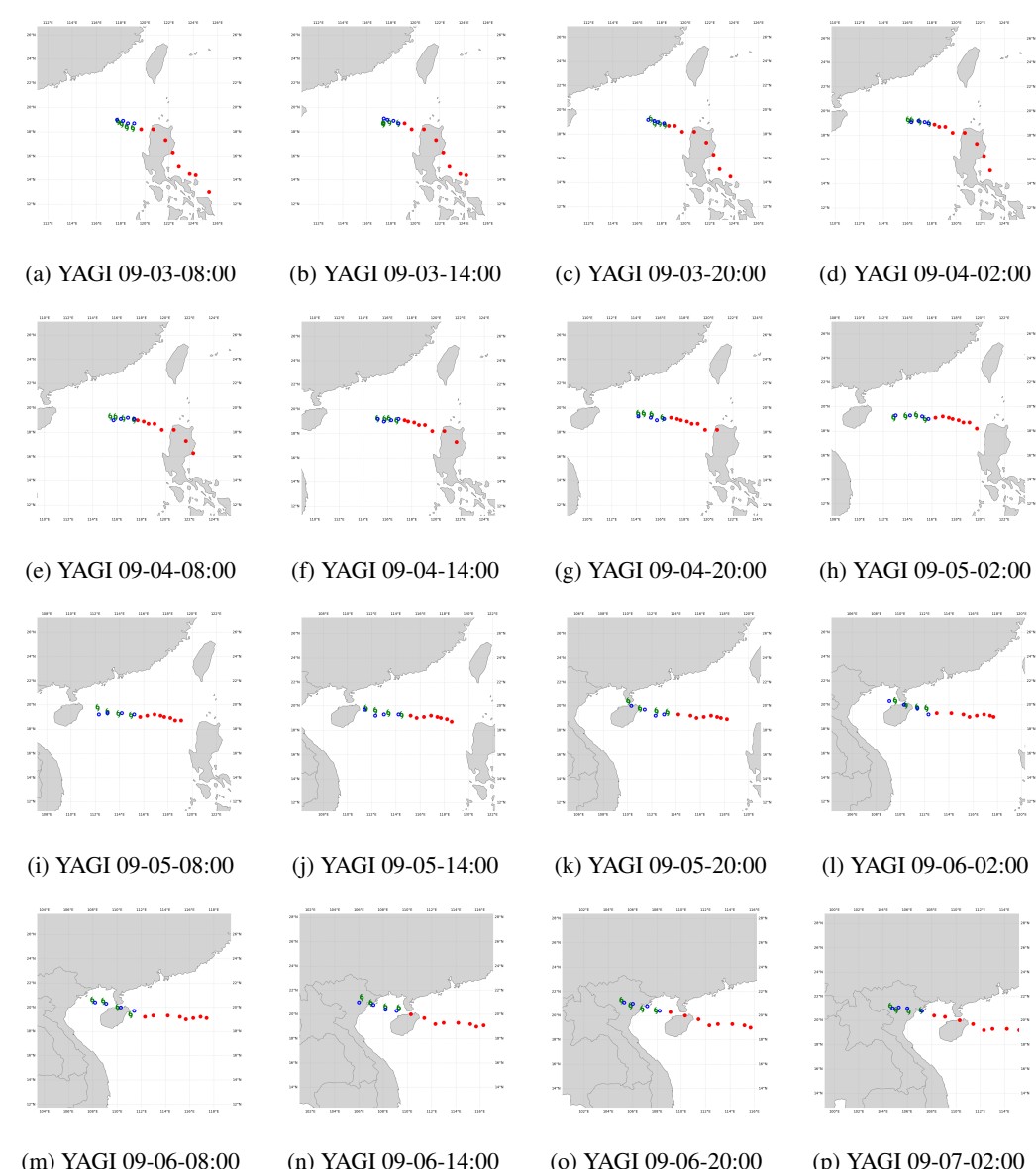

(a) YAGI 09-03-08:00    (b) YAGI 09-03-14:00    (c) YAGI 09-03-20:00    (d) YAGI 09-04-02:00

(e) YAGI 09-04-08:00    (f) YAGI 09-04-14:00    (g) YAGI 09-04-20:00    (h) YAGI 09-05-02:00

(i) YAGI 09-05-08:00    (j) YAGI 09-05-14:00    (k) YAGI 09-05-20:00    (l) YAGI 09-06-02:00

(m) YAGI 09-06-08:00    (n) YAGI 09-06-14:00    (o) YAGI 09-06-20:00    (p) YAGI 09-07-02:00

Figure 6: Results showcase trajectory predictions using AOT-TCNet for the TC YAGI. The red dots are input, the green dots are prediction, and the blue dots are GT.

process offshore. The subsequent northward turn and landfall of GEIMI in the Yilan area resulted from the combined effects of multi-scale systems: strong convergence brought by the northerly jet, channeled flow in the Taiwan Strait, and southwesterly flows in the southwestern quadrant of the TC overlapped, triggering intense convective activity. The resulting wrapping rainbands pushed the entire circulation eastward. Meanwhile, the northerly jet gradually weakened and dissipated due to topographic friction, allowing the large-scale steering flow to regain dominance over the TC's movement. Combined with stronger winds on the eastern side of the TC and structural asymmetry, these factors collectively led to the northward turn after looping. In addition, the lee-side low, acting as a secondary center, interacted with the primary circulation through the Fujiwhara effect, further influencing the looping path.

*Discussion and Future Work*: Firstly, the coarse resolution of the data used to train the models restricts them to estimating general impact ranges or approximate locations, rather than providing precise predictions. Secondly, the limited scope and granularity of the training datasets reduce

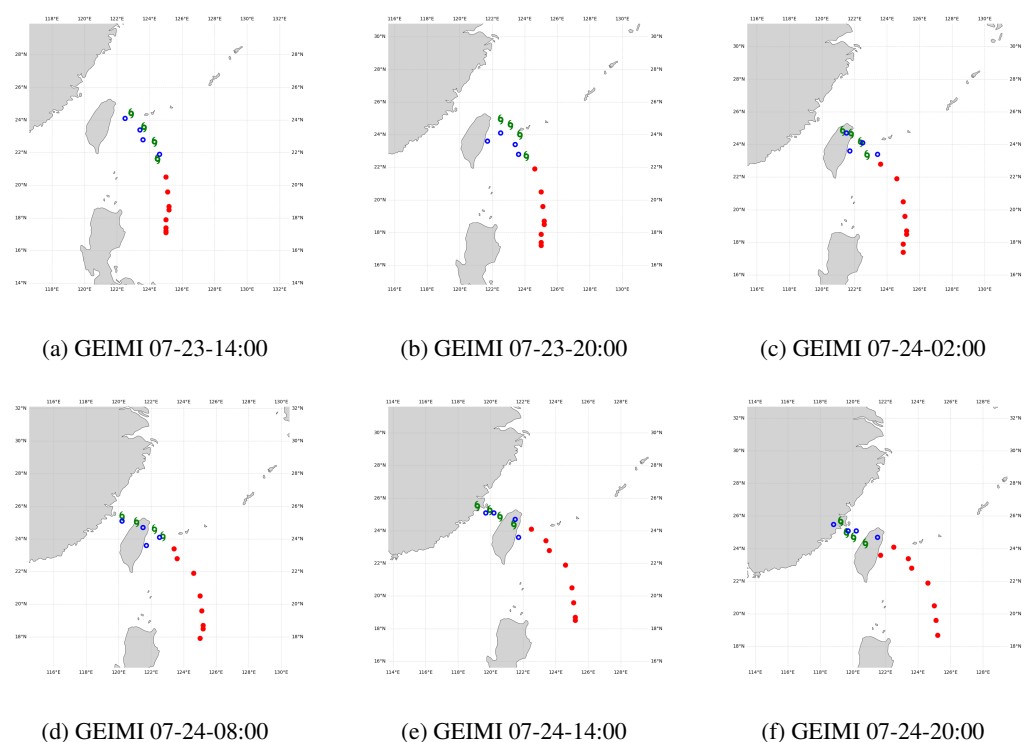

Figure 7: Results showcase trajectory predictions using AOT-TCNet for the TC GEIMI. The red dots are input, the green dots are prediction, and the blue dots are GT.

inter-model diversity and weaken regional predictive capability. Experiments have shown that such limitations can lead to underestimations of extreme events and TC intensity, particularly under the increasing influence of global warming, which is reflected in the decline in predictive performance over the past two years. Thirdly, TCs are significantly influenced by large-scale atmospheric circulation patterns over extended time scales, and deep learning models struggle to capture such long-term dependencies, resulting in increased forecast errors at longer lead times. To address these issues, in future work, we will use diffusion models to reconstruct TC data with high temporal and spatial resolution. Additionally, applying temporal models to atmospheric data is expected to improve long-term forecasting capabilities.

## A.9 DECLARATION OF LARGE LANGUAGE MODEL USAGE

While writing this paper, the authors used a LLM to polish the wording for enhanced readability, with the LLM making no substantive contributions. The core ideas, theoretical derivations, experimental design, result analysis, and conclusions proposed in this paper represent solely the work and views of the authors. The authors assume full responsibility for all content of this paper.

