# OpenReview forum: "Atmosphere-Ocean-Terrain Coupled Ensemble Forecast Model Improves Forecasts of Abnormally Deflecting Tropical Cyclones"
_ICLR.cc/2026/Conference — ICLR 2026 Conference Withdrawn Submission_

### Official Review · Reviewer_dh1G · 2025-10-27

**Soundness:** 2
**Presentation:** 2
**Contribution:** 2
**Rating:** 4
**Confidence:** 4

**Summary:**

The paper presents AOT-TCNet, a multimodal deep-learning framework for tropical cyclone (TC) forecasting that explicitly couples atmosphere, ocean, and terrain data. The authors (1) construct AOT-TCs, a large multimodal dataset for the Western North Pacific (1950–2024) combining CMA-BST tracks, ERA5 atmospheric fields, CODCv1 ocean observations, and GEBCO bathymetry; and (2) propose a modeling pipeline combining specialized encoders, a TC-Mode-Adaptive Mixture-of-Experts (TMA-MoE) to capture multimodal trajectory patterns, and Reinforcement Learning with Physical Feedback (RLPF) to encourage physically consistent and diverse expert outputs. Experiments (2017–2024/2020–2024 splits reported) show improvements over several deep-learning baselines and competitive results versus operational NWP systems on short lead times, with particular gains reported for “abnormally deflected” TCs. Code & data (anonymous) are claimed to be released.

**Strengths:**

1. Important problem & strong motivation. Forecasting anomalous / abruptly deflected TCs is societally important; connecting terrain/ocean/atmosphere information is well motivated.
2.Large multimodal dataset. The AOT-TCs dataset (1950–2024) integrating multiple public sources (CMA-BST, ERA5, CODCv1, GEBCO) is a valuable resource and likely to help future work. The paper documents the data sources and variables used.
3.Novel modeling combination. Combining MoE (to learn multimodal track distributions) with an RL-style physical feedback loop is an interesting design to promote diversity and physical consistency.
4.Empirical gains on short lead times and abnormal cases. Reported improvements for 6–18h forecasts, and specialized ablation showing benefit from heterogeneous inputs and MoE, indicate the approach has practical merit

**Weaknesses:**

1.RLPF (RL) training details insufficient and ablation of RL missing.
Although RLPF is presented as a core contribution, the paper lacks details of its implementation: the policy optimization method, reward formulation, training schedule, and stability control are not described. Without this information, the reproducibility and interpretability of the claimed physical benefits are limited. Provide an ablation that isolates the contribution of RLPF

2.Ablation of terrain and other modality contributions missing.
The manuscript claims that terrain information is critical for abnormal deflection cases but provides no quantitative isolation of this factor. Controlled experiments that remove or swap the terrain input are needed to validate this claim.

3.Lack of uncertainty quantification.
Reported results are given as mean errors without confidence intervals or statistical testing. Given the stochastic nature of tropical cyclone behavior, the paper should provide uncertainty estimates (e.g., standard deviation or bootstrap CI) to confirm that improvements are statistically significant.

4.Train / test split protocol unclear and may risk temporal leakage.
The manuscript does not explicitly state how the training, validation, and test sets are separated across years. Given the strong temporal autocorrelation in tropical cyclone data, improper splitting could lead to data leakage and inflated performance. The authors should clearly provide the chronological split strategy and ensure that no future data are used in training.

**Questions:**

1.How exactly are the training, validation, and test periods defined? Are they separated chronologically by year to prevent temporal leakage? Please provide a clear table of year ranges for each split.

2.Provide an ablation that removes RLPF but keeps the MoE + supervised losses. How much of the improvement (especially on abnormal cases) remains?

3.How sensitive are results to the number of MoE experts, the gating network capacity, and the σ hyperparameter in Eq. (4)? Please report.

4. Could you provide results of an ablation removing terrain data or replacing it with a constant field, to verify that terrain features indeed contribute to improved forecasts of abnormal deflections?

---

### Official Review · Reviewer_sBAM · 2025-10-27

**Soundness:** 3
**Presentation:** 2
**Contribution:** 3
**Rating:** 4
**Confidence:** 4

**Summary:**

This paper introduces AOT-TCNet, a deep learning framework for tropical cyclone (TC) forecasting that explicitly couples atmospheric, oceanic, and terrain processes. The authors also present AOT-TCs, the most comprehensive multimodal dataset for the Northwest Pacific (1950–2024), integrating atmospheric, oceanic, and topographic variables. The model combines a TC-Mode-Adaptive Mixture of Experts (TMA-MoE) with a Reinforcement Learning with Physical Feedback (RLPF) strategy to enforce physical consistency. Empirical evaluations across TC events from 2017–2024 demonstrate significant improvements over both deep learning and operational numerical weather prediction (NWP) systems, especially in forecasting abnormally deflected TCs. Ablation studies and comparative analyses confirm the contribution of each architectural component and dataset modality.

**Strengths:**

The explicit coupling of atmosphere, ocean, and terrain domains via multimodal embeddings is novel and physically justified. Experiments are comprehensive, including global operational comparisons and ablation studies. The ability to accurately forecast anomalously deflected TCs (e.g., GEIMI, GAEMI, YAGI) represents a major advancement for real-world disaster prediction systems.

**Weaknesses:**

1. While the dataset represents a major and valuable contribution of this work, its presentation is largely confined to the appendix. Presenting at least a concise summary in the main body would help readers better appreciate the dataset’s scale and significance, rather than discovering these crucial details only in the supplementary section.

2. While results are impressive in the Northwest Pacific, the model’s transferability to other basins (e.g., Atlantic, Indian Ocean) is not evaluated.

3. Although physical consistency is enforced via RLPF, the paper provides little qualitative analysis (e.g., feature attribution, explainability) of how physical variables contribute to predictions.

4. The paper emphasizes efficiency but omits precise training/inference time comparisons versus NWP baselines.

**Questions:**

1. Could the authors provide a concise summary table or visualization in the main text (e.g., number of samples, variable categories, spatial/temporal resolution)?

2. Have the authors conducted any preliminary experiments or transfer tests on other regions such as the North Atlantic or Indian Ocean to assess the model’s adaptability? Would fine-tuning with limited regional data be sufficient for generalization, or would re-training from scratch be necessary?

3. Could the authors provide qualitative or quantitative evidence (e.g., sensitivity analysis, feature attribution, or ablation by variable type) to clarify which components—terrain, SST, or atmospheric fields—most affect deflection prediction?

4. Could the authors report the average training time, inference time per forecast, and resource utilization (GPU type, memory, FLOPs) to substantiate the efficiency claim?

---

### Official Review · Reviewer_1auv · 2025-10-29

**Soundness:** 2
**Presentation:** 1
**Contribution:** 3
**Rating:** 4
**Confidence:** 3

**Summary:**

The authors study the task of tropical cyclone forecasting, where the active challenges with the traditional TC forecasting models are capturing the heterogeneity and multimodality of the data, having high computational cost, and the issue of catastrophic forgetting, where the model drifts towards newly learned distributions and forget about the previous TC sub-patterns that it has learned. Hence, the authors propose AOT-TCNet, which is a multimodel TC forecasting model that combines a Mixture of Experts (MoE) architecture with a Reinforcement Learning with Physical Feedback (RLPF) training strategy. The authors also provide a novel dataset called AOT-TCs for tropical cyclone forecasting.

**Strengths:**

- The authors propose a Mixture of Experts model trained using an RLHF framework. Combining reinforcement learning with ensemble learning is an interesting direction, particularly in the context of weather forecasting. To the best of my knowledge, the proposed architecture and problem setting are both novel and interesting.
- The authors not only propose a novel model architecture but also introduce a multimodal, multi-source dataset named AOT-TCs for tropical cyclone forecasting, which is a valuable contribution to the literature.
- The code and dataset is provided by the authors, and at first glance, seems to satisfy what they have written in their manuscript.
- The experiments cover both short-term (6h) and medium-term (24h) forecasting across different years and targets, including distance, pressure, and wind speed. The proposed architecture is compared against numerous baseline methods with varying designs and strengths, representing a broad range of approaches. The visualizations of predictions, both in this section and in the Appendix, further enhance the value of the experimental analysis.

**Weaknesses:**

- The discussion on combining reinforcement learning with MoE systems should be expanded in Section 2. The authors state that “MoE systems and reinforcement learning have been proven effective in enhancing reasoning capabilities”, but provide no examples from the literature. Since the proposed model integrates an MoE based architecture with RLHF, this topic deserves a more extensive review.
- In Figure 2, the architecture of AOT-TCNet (particularly the green section) is difficult to interpret. According to the Appendix, the TC-Encoder uses LSTMs, while the AOT-Encoder uses 3D-UNets. It is unclear how the Feature Extractor is related to the 3D-UNet and how DATA is connected to the LSTM in the figure, as their placement below those components suggests some relation. Additionally, the figure seems to indicate that the LSTM output is fed into the 3D-UNet, an MLP, and another MLP, which adds to the confusion. The figure should be clarified to better illustrate these relationships.
- The authors state that “the model samples or selects a expert from $\pi$ and predicts a trajectory…”. which seems confusing. From the rest of the paper, I understand that the gating network learns to select the expert that generates the trajectory pattern most consistent with the current TC path, but the selection process is unclear. Does the gating network “sample” expert $E_i$ with probability $\pi_i$ or does it select the expert with the highest probability? From Figure 2 (red box), it can be understood as for some cases the expert with the highest probability is chosen and multiple experts can contribute for other cases. Is that what should be understood?
- Section 3.3 describes the RLHF system implemented on top of the MoE architecture, but this part is somewhat difficult to follow. In Equation 8, shouldn’t $r_{phy,i}$ be independent of $i$ based on the provided formulation? Also, the term $Y^{atm}$ is not clearly defined. In Equation 9, it seems the correct notation should be $r_{mode,i}$ instead of $r_{mode}$. Overall, this subsection would benefit from a clearer and more precise explanation.

**Questions:**

- Please see the weaknesses section. Overall, the proposed architecture and dataset are interesting, but the manuscript is difficult to follow and lacks clarity. The writing could be significantly improved, as there are also several grammatical errors throughout the paper.
- The authors begin the abstract by emphasizing “lower computational cost and faster operation speed”. In that case, shouldn’t they provide a comparison of training times across models, or at least for their own setup?

---

### Note · Authors · 2025-11-25

I have read and agree with the venue's withdrawal policy on behalf of myself and my co-authors.